# Effects of Different Diets on Biological Characteristics of Predatory Mite *Amblyseius Eharai* (Acari: Phytoseiidae)

**DOI:** 10.3390/insects14060519

**Published:** 2023-06-03

**Authors:** Nguyen T. P. Thao, Nguyen T. Thuy, Ho L. Quyen

**Affiliations:** 1Institute of Tropical Biology, Vietnam Academy of Science and Technology, Ho Chi Minh City 700000, Vietnam; thuynguyen0224@gmail.com; 2Plant Quarantine Subdepartment Zone II, Plant Protection Department, Ministry of Agriculture and Rural Development, Ho Chi Minh City 700000, Vietnam; quyentpv@gmail.com

**Keywords:** *Amblyseius ehrai*, artificial diet, *Artemia fanciscana*, *Panonychus citri*, *Tetranychus urticae*, life table

## Abstract

**Simple Summary:**

In this study, we investigated the effects of different diets on the development and reproduction of *Amblyseius eharai* (Amitai and Swirskii) (Acari: Phytoseiidae), a predatory mite used for biological control. The results show that feeding on *Panonychus citri* (McGregor) (Acari: Tetranychidae) resulted in the fastest life cycle completion time, the longest oviposition period, and the highest number of eggs laid per day. Feeding on *Tetranychus urticae* (Koch) (Acari: Tetranychidae) and pollen resulted in similar life cycle completion times, while feeding on Artemia cysts and an artificial diet resulted in longer life cycle completion times and shorter oviposition periods. These findings provide insights into the optimization of the diets of *A. eharai* for effective use for biological control.

**Abstract:**

In this study, we investigated the effects of different diets on the development and reproduction of the predatory mite *Amblyseius eharai*. The results show that feeding on citrus red mite (*Panonychus citri*) led to the fastest life cycle completion (6.9 ± 0.22 days), the longest oviposition period (26.19 ± 0.46 days), the greatest female longevity (42.03 ± 0.43 days), and the highest total number of eggs per female (45.63 ± 0.94 eggs). Feeding on *Artemia fanciscana* cysts resulted in the highest oviposition rate (1.98 ± 0.04 eggs), a high total number of eggs per female (33.93 ± 0.36 eggs), and the highest intrinsic rate of increase (r_m_ = 0.242). The hatching rate did not differ significantly among the five types of food, and the proportion of females ranged from 60% to 65% across all diets.

## 1. Introduction

The predatory mite of the family Phytoseiidae is an important biological control agent for various pests in greenhouses and net houses worldwide [1]. Therefore, many studies have been conducted on the rearing of phytoseiid mites using various types of food sources [2,3,4,5,6,7,8,9,10,11,12,13]. *Amblyseius eharai* (Amitai and Swirskii) (Acari: Phytoseiidae) is a type III-b generalist predator with a wide prey spectrum and is capable of developing on pollen [7,14]. In the larval stage, the family Phytoseiidae can be divided into three groups based on their feeding characteristics: a group that must feed on larva (OFL), a group that does not feed on larva (NFL), and a group for which feeding on larva is facultative (FFL). *A. eharai* belongs to the OFL group [15]. *Amblyseius eharai* is a native predator and usually appears at the beginning of the apple-growing season in Korea [16]. *A. eharai* has also been reported in Japan, China, Taiwan, and Malaysia [17,18] and is considered as a biological control agent for *Aceria litchii* (Keifer) (Acari: Eriophyidae), *Calepitrimerus vitis* (Nalepa) (Acari: Erio phyidae), *Panonychus citri* (McGregor) (Acari: Tetranychidae), *Pseudo dendrothrips mori* (Niwa) (Thysanoptera: Thripidae), *Tetranychus kanzawai* (Kishida) (Acari: Tetranychidae), and *Tetranychus urticae* (Koch) (Acari: Tetranychidae) [16,19,20,21,22]. Previously, *A. eharai* was misidentified as several other predatory mite species, such as *Amblyseius newsami* (Evans) and *Amblyseius cantonensis* (Schicha) (Acari: Phytoseiidae), until the 1980s [22]. Later, Amitai and Swirski distinguished and classified *A. eharai* from these species based on the difference in the length of the dorsal shield [23]. *A. eharai* is known to be one of the most common biological control agents for *Panonychus citri,* which is a main pest of citrus trees [15,24,25,26,27]. *A. eharai* is also known as a biological control agent for plant-feeding mites on apple trees at the beginning of the season [16]. *A. eharai* has been recorded on various types of plants with different leaf shapes, such as conifers, shrubs, herbs, and vines [28] and on various crops, such as tea, longan, ornamental plants, bamboo, Japanese medlar, and cotton [19]. *A. eharai* is the most common predatory mite species in Korea [28] and has a broad prey spectrum, including thrips, aphids, and other harmful mites [13,15]. Many studies have been conducted on the use of different prey and pollen to rear *A. eharai*, such as studies by Saito and Mori (1975) [1], Wei et al. (1993) [29], and Pu et al. (1995) [30]; however, not all these studies achieved optimal results, owing to the difficulty in collecting and preserving pollen [31]. *A. eharai* is commonly found in citrus orchards in the southern provinces of Vietnam. In the present study, samples were collected and reared in the laboratory. Preliminary results show that *A. eharai* is suitable for rearing under laboratory conditions. Therefore, we conducted a study on mass rearing of *A. eharai* for use in the development of an IPM model for citrus orchards in southern Vietnam.

### Materials and Methods

The experiment was conducted in the insect laboratory (25 ± 1 °C and 75 ± 5% RH) of the Department of Animal Cell Technology, Institute of Tropical Biology. The study was carried out from March 2021 to June 2022.

## 2. Materials

### 2.1. Predatory Mite Source

Predatory mites were collected in citrus orchards in the southwestern provinces of Vietnam and brought back to the laboratory for sorting and rearing purposes. 

The predator-rearing unit included bean leaves resting upside down on water-saturated cotton wool in a plastic Petri dish (90 × 15 mm) on a water-saturated sponge in a plastic container (26 cm× 16 cm× 7 cm). Water was added to the rearing containers when necessary to keep the leaves fresh and prevent predatory mites and mites from escaping. Every 2 days, *T. urticae*-infested rose leaves were added to the rearing unit. This culture was started 3 months before the beginning of the experiments. The rearing unit was maintained at 25 ± 1 °C and 75 ± 5% RH with a 16 L: 8D hour photoperiod in a growth chamber.

### 2.2. Prey and Alternative Food Source

*Tetranychus urticae* and *Panonychus citri* were collected from citrus trees and reared on kidney bean plants (*Phaseolus vulgaris*) grown in beds under commercial conditions (Cocopeat: Perlit; 60%: 40%) in greenhouses at the Institute of Tropical Biology.

The pollen source (pollen from citrus trees) was collected in a citrus orchard in the south of Vietnam, frozen at −18 °C, thawed before experiments, and kept in a refrigerator at 5 °C.

*Artemia franciscana* (Kellogg) (Branchiopoda: Artemiidae) cysts were provided by HTX ARTEMIA (Ho Chi Minh, Viet Nam) and originated from Vinh Chau (Bac Lieu province, Viet Nam); the cysts were brought to the laboratory and treated under ultraviolet light to kill the embryos; then, the outer shells were removed. The cysts were then stored in a refrigerator at 5 °C and used within one week.

Five types of food were tested to determine the appropriate diet for the mass rearing of *A. eharai*: larval-stage *T. urticae*, larval-stage *P. citri*, *Artermia franciscana* cysts, pollen citrus, and an ArP artificial diet (16.6% sucrose, 16.6% tryptone, 16.6% yeast extract, 6.7% fructose, 16.6% egg yolk powder, 0.13% vitamin mix, and 20% dry-ground *Artermia franciscana* cysts) [32].

## 3. Methods

### 3.1. Effects of Different Diets on the Development of Amblyseius eharai

Fifty separate rearing units were prepared, and an egg of *A. eharai* was placed in each unit. 

Each type of food was provided to ten rearing units. The units were checked daily, and the growth and development time of *A. eharai* were recorded until the adult stage. The rearing units were cleaned and provided with food every two days.

Monitoring criteria: Observations were made every 12 h, and the development time of predatory mites (egg, larva, protonymph, and deutonymph stages and of all the juvenile phase (egg–adult)) were recorded for five different types of food.

Experimental design: The experiment was arranged in a completely randomized design with five treatments and three replications. The experimental layout was as follows Table 1.

### 3.2. Effects of Different Diets on the Reproduction Ability of Amblyseius eharai

The predatory mites were fed on various food sources until the final molt stage during which they became adults, then separated into individual Petri dishes. Each predatory mite was allowed to mate with a male adult for 2 days while continuing to be fed with the same food as during their developmental stages. The rearing units were checked daily, and the time of preoviposition, oviposition, and post-oviposition periods, as well as the fecundity and oviposition rate, were recorded.

#### 3.2.1. Life Table Parameters

The intrinsic rate of increase (r_m_) was calculated according to the formula proposed by Birch (1948) [33].
$$\sum l_{x}m_{x}e^{-r_{m^{*}}x}=1$$
where x equals the female age (days), l_x_ is the age-specific survival of females at age x, and m_x_ is the number of daughters produced per female at age x. The latter parameter is obtained by multiplying the mean number of eggs laid per female by the proportion of female offspring produced at age x. 

The net reproductive rate (R_0_), i.e., the mean number of female offspring produced per female (females/female) is expressed as:$$R_{0}=\underset{\cdot }{\sum }I_{x}m_{x}$$

The mean generation time (T) was defined as the length of time that a population needed to increase to R_0_-fold of its size (i.e., $e^{r_{m\;}T}$ = R_0_ or $\lambda^{T}$ = R_0_) at the stable age–stage distribution. The mean generation time is calculated as:$$T=(\ln\;R_{0})/r_{m}$$

The finite rate of increase (λ) is expressed as:$$\lambda =e^{r_{m}}$$

#### 3.2.2. Statistical Analysis

One-way analysis of variance (ANOVA) (SigmaPlot 11.0) was conducted to evaluate the effects of diet on the immature stages, preoviposition, oviposition, and post-oviposition period daily, as well as the total oviposition and adult longevity, of *A. eharai* using a Tukey test; if a Levene test indicated heteroscedasticity, a rank test was used instead of a Tukey test. In all tests, *p* values less than or equal to 0.05 were considered significant.

## 4. Results

### 4.1. Effects of Different Diets on the Development of Amblyseius eharai

Table 2 shows that when feeding on *P. citri*, *A. eharai* completed its life cycle the fastest (6.9 ± 0.22 days); this difference was statistically significant compared to feeding on the other four types of food. When feeding on *T. urticae*, the life cycle of *A. eharai* (8.2 ± 0.13 days) did not differ significantly from that when feeding on pollen (8.4 ± 0.61 days).

The longest life cycle completion time for *A. eharai* was recorded when feeding on Artemia cysts (9.57 ± 0.19 days); this difference was not statistically significant compared to feeding on ArP (9.23 ± 0.21 days). The life cycle of *N. barkeri* on *T. urticae* was reported to be 8.4 days at 25 °C [34], which is equivalent to *A. eharai* feeding on *T. urticae* in this study. However, the life cycle of *A. eharai* feeding on *T. urticae* in this study was longer than that of some other small predatory mites also fed on *T. urticae*, specifically *N. barkeri* (6.3 days) [35] and *N. californicus* on *T. urticae* (5.09 days) [36]. In a study by Park and Lee (2020), it was observed that as the temperature increased, the life cycle of *A. eharai* became shorter when feeding on *T. urticae*, and at 21.6 °C, *A. eharai* completed its life cycle in approximately 8.8 days, which is similar to the findings of this study.

In a study by Ji et al. [24] (2013), the life cycle of *A. eharai* at 25 °C when feeding on *P. citri* larvae was reported to be 9.7 days, and that on *P. citri* eggs was 7.7 days, which is higher than the length of life cycle reported in this study (6.9 ± 0.7 days).

The use of pine pollen was found to be effective as a substitute food for the development of *A. eharai* populations [37]. DNA analysis also revealed that *A. eharai* uses two types of pollen from *Digitaria ciliaris* (Retz.) Koeler (Poales: Poaceae) and *Pinus densiflora* Siebold and Zucc. (Pinales: Pinaceae) as additional food sources [38].

The females of *A. eharai* have different colors depending on their diets. When they feed on ArP and pollen, they have a lighter and more transparent coloration (Figure 1A). When they feed on *P. citri* and *T. urticae,* their coloration becomes darker (Figure 1B). Figure 2 illustrates the mating process of *A. eharai.*

### 4.2. Effects of Different Diets on the Reproduction Ability of Amblyseius eharai

The results presented in Table 3 show that the longest oviposition period when feeding on prey was recorded for *P. citri* (26.19 ± 0.46 days); the difference was statistically significant compared to the other four types of food. The oviposition period of *A. eharai* when feeding on Artemia cysts (18.03 ± 0.22 days) was shorter than that when feeding on its two natural prey, *P. citri* and *T. urticae*, but still significantly longer than that when feeding on pollen (12.07 ± 0.35 days) and ArP (13.13 ± 0.36 days). All these differences were statistically significant. 

The lifespan of *A. eharai* females was highest when feeding on *P. citri* (42.03 ± 0.43 days), and lowest when feeding on pollen (28.77 ± 0.24 days). There was a statistically significant difference in lifespan across all diets. Among the three alternative food sources, the lifespan of *A. eharai* offspring was highest when consuming artemia cysts (33.93 ± 0.36 days).

On average, the highest number of eggs laid per day was achieved when feeding on Artemia cysts (1.98 ± 0.04 eggs/day/female); the difference was statistically significant compared to the other four types of food. The lowest number was observed when feeding on ArP (1.48 ± 0.05 eggs/day/female).

The total number of eggs/female was highest when feeding on the two natural prey species, *P. citri* (45.63 ± 0.94 eggs) and *T. urticae* (40.29 ± 0.83 eggs), with the highest number achieved when feeding on *P. citri*; the difference was statistically significant compared to the other three types of food. When feeding on Artemia cysts, the total number of eggs/female was also quite high (35.87 ± 0.48 eggs/female).

The hatching rate did not differ significantly among the five types of food over the course of the year, with the highest rate observed when feeding on *T. urticae* (81.47 ± 0.71%), followed by Artemia cysts (80.72 ± 2.07%) and *P. citri* (78.38 ± 0.83%).

The highest proportion of females was observed when feeding on pollen (65.27 ± 1.75%), with no statistically significant differences observed among the other feeding regimes. The proportion of females ranged from 60% to 65% across all diets.

The results presented in Table 4 show the highest net reproductive rate (R_0_) and generation time (T) when feeding on *P. citri* (R_0_ = 23.06 and T = 14.21) and the highest intrinsic rate of increase (r_m_) when feeding on artemia cysts (r_m_ = 0.242).

## 5. Discussion

In this study, we evaluated various aspects of the biological parameters and life history traits of *A. eharai* under different dietary conditions. Our findings shed light on the effects of the diets on the lifespan and population growth of this predatory mite. 

In this study results demonstrated that when *Amblyseius eharai* was fed on natural prey sources (*P. citri* and *T. urticae*), it exhibited higher indicators of oviposition period, total number of eggs, and female longevity compared to when it was fed on alternative food sources (artemia cysts, pollen, and ArP). However, the index of oviposition rate was highest when the mites were fed on Artemia cysts, with a rate of 1.98 ± 0.04 eggs per female per day (Table 3).

When *A. eharai* was fed on Artemia cysts, it showed the highest lifespan and total number of eggs among the three alternative diets. The scores were relatively high when fed on *T. urticae* as well. Specifically, when fed on Artemia cysts, the lifespan and total number of eggs for *A. eharai* were 33.93 days and 35.87 eggs, respectively. When fed on *T. urticae*, the corresponding values were 35.23 days and 40.29 eggs, respectively.

Regarding the hatching egg proportion of the progeny and the female proportion of the progeny, the values were similar across all diets and exceeded 70% for hatching egg proportion of the progeny and 60% for female proportion of the progeny.

The results of this study align with certain findings from other studies.

The oviposition period of *A. eharai* when feeding on *T. urticae* decreased as the environmental temperature increased, with oviposition periods of 22.85 days and 19.65 days at 21.6 °C and 24.1 °C, respectively [39], which is similar to the oviposition period of *A. eharai* feeding on *T. urticae* in this study. At 33.2 °C, *A. eharai* does not lay eggs [39]. At 25 °C, when feeding on *P. citri*, the oviposition period of *A. eharai* is 26.3 ± 2.2 days [24], similar to the oviposition period of *A. eharai* reported in this study. The oviposition period of *Amblyseius swirskii* when feeding on *P. citri* is 25.8 ± 2.0 days [24], which is shorter than that of *A. eharai.*

The lifespan of *A. eharai* when feeding on *P. citri* (42.03 ± 1.36 days) is similar to that reported in a study by Ji et al. [24] (2013) when *A. eharai* was fed on *P. citri* at 25 °C (43.5 ± 0.7 days).

The lifespan of *A. eharai* feeding on *T. urticae* decreases as the temperature increases, with the longest lifespan observed at 18 °C (30.04 ± 2.725 days) and the shortest at 33.2 °C (3.73 ± 0.709 days) [39]. When fed on *T. urticae* in this study, the female *A. eharai* lifespan (35.23 ± 1.10 days) was higher than that of *E. finlandicus* feeding on red spider mite (23.7 ± 1.41 days) and *T. pyri* (35.6 ± 1.80 days) [21].

When feeding on ArP, the lifespan of *A. eharai* in this study (30.97 ± 0.94 days) was similar to the lifespan of *A. swirskii* females feeding on ArP in a study by Nguyen Duc Tung (30.75 ± 0.83 days) [32].

The average daily egg production of *A. eharai* when feeding on *T. urticae* increases as the temperature fluctuates from 18 °C to 27.4 °C, then decreases at 30.2 °C [39]. In this study, at 24.1 °C, the average daily egg production (1.73 eggs/female/day) was similar to that of *A. eharai* feeding on *P. citri* (1.74 ± 0.04 eggs/female/day). *A. eharai* feeding on *T. urticae* has an oviposition rate of 1.89 ± 0.12 eggs/female/day, which is similar to that of A. swirskii feeding on *T. urticae* (1.8 ± 0.1 eggs/female/day) [40] and much higher than that of some other predatory mite species feeding on the same prey, such as *E. finlandicus* and *T. pyri* (0.64 ± 0.05 eggs/female/day and 0.80 ± 0.09 eggs/female/day, respectively) [41].

The oviposition rate of *A. eharai* when feeding on *P. citri* in this study (1.74 ± 0.04 eggs/female/day) was much lower than in a study by Ji et al. [24] (2013) (2.59 ± 0.03 eggs/female/day).

When feeding on *T. urticae*, the total number of eggs laid by *E. finlandicus* and *T. pyri* (12.2 ± 0.92 and 19.9 ± 0.93, respectively) was much lower than the total number of eggs laid by *A. eharai* when feeding on *T. urticae* in this study (40.29 ± 0.83 eggs), which is similar to that of *A. eharai* feeding on *T. urticae* at 24.1 °C (42.31 ± 2.731 eggs) [39].

The total number of eggs/female *of A. eharai* when feeding on *P. citri* in this study was much lower than that reported in a study by Ji et al. [24] (2013) (67.70 ± 1.32 eggs/female).

The hatching egg proportion of the progeny of *A. eharai* when feeding on an ArP diet (78%) is similar to that of *A. swirskii* when feeding on the same ArP diet (78%) at 23 °C [32].

When feeding on *P. citri* and *T. urticae, A. eharai* had high levels of biological indicators assessing the ability of the population to develop (R_0_, T, r_m_, and λ). Notably, on food, Artemia eggs also had the highest natural gain among the five foods (higher than on food that was natural prey). Among the three alternative feeds, the indices (R_0_, T, r_m_, and λ) were highest on a diet of Artemia cysts, which shows that the use of Artemia cysts in *A. eharai* biomass culture is appropriate.

The intrinsic rate of increase (r_m_) when feeding on prey *P. citri* at 25 °C reported in this study (r_m_ = 0.221) is higher than that reported in a study by Ji et al. [24] (2013) (r_m_ = 0.1711).

Reports on the life table and biological parameters of *A. eharai* on different food sources are still limited. Saito and Mori [35] (1981) reported an intrinsic rate of increase of *A. eharai* (*Amblyseius deleoni* (Muma and Denmark), which is another name for *A. eharai*) when fed on T. urticae to be 0.286 at 25 °C, which is higher than that reported in this study (0.226). Moreover, the intrinsic rate of increase (r_m_) of *Amblyseius eharai* reported in this study is also higher than that of some other predatory mites on the same food source, *T. urticae*, including *Typhlodromus pyri* (r_m_ = 0.11 at 25 °C) *Euseius finlandicus* (r_m_ = 0.09 at 25 °C) [30], *Euseius finlandicucs* (r_m_ = 0.11) [1], *N. barkeri* (r_m_ = 0.221 at 25 °C) [11], and *C. negevi* (r_m_ = 0.16 at 27 °C) [42].

According to Park and Lee [39] (2020), when testing *A. eharai* at different rearing temperatures, the highest intrinsic rate of increase (0.2619) was recorded at 27.4 °C, which is similar to the result of *A. eharai* feeding on Artemia cysts (r_m_ = 0.242) reported in this study; the lowest rate (0.0792) was recorded at 18 °C when feeding on *T. urticae*. When feeding on *Carpoglyphus lactis* (Linnaeus) (Acari: Carpoglyphidae) at 25 ± 1 °C, the intrinsic rate of increase was 0.253 [33].

The intrinsic rate of increase of *A. eharai* when feeding on *Pinus thunbergii* pollen is 0.216 [41], which is higher than the intrinsic rate of increase of *A. eharai* when feeding on citrus tree pollen reported in this study (r_m_ = 0.215). However, the intrinsic rate of increase of *A. eharai* when feeding on pollen is higher than that of some other predatory mites feeding on different pollen species suitable for each species: *Euseius finlandicus* feeding on *Brassica napus* L. pollen (r = m = 0.124 at 25 °C) [30], *Typhlodromus pyri* feeding on *Brassica napus* L. pollen (r_m_ = 0.05 at 25 °C) [30], and *Euseius finlandicucs* feeding on *Tulipa gesnerana* L. pollen (r = 0.101 at 25 °C) [43].

The intrinsic rate of increase of *A. eharai* feeding on ArP in this study (r_m_ = 0.196) was higher than the correlation coefficient of *A. swirskii* on the same substrate in Nguyen Duc Tung’s study (r_m_ = 0.159 at 23 °C) [32] and higher than that feeding on other small predatory mite species raised on the same artificial diet formula, including *Neoseiulus cucumeris* (r_m_ = 0.090 at 25 °C), *Amblyseius andersoni* (r_m_ = 0.144 at 25 °C), and *Amblydromalus limonicus* (r_m_ = 0.212 at 25 °C) [36].

This study provides valuable insights into the biological parameters and life history traits of *A. eharai* under different dietary conditions. The findings contribute to our understanding of the population dynamics and reproductive performance of this predatory mite species. Further studies are necessary to explore additional factors that may impact the biological parameters of *A. eharai*, allowing for more comprehensive knowledge of this species and its potential applications in pest control strategies. It also highlights the role of Artemia cysts as a potential alternative food source in *A. eharai* biomass culture.

## 6. Conclusions

This research reveals that *A. eharai* exhibits favorable fertility, longevity, and the ability to establish populations using two types of prey, namely *T. urticae* and *P. citri*. This suggests that *A. eharai* can be utilized as a management tool for controlling these two pests. Additionally, A. eharai demonstrated the capability on alternative food sources, including artemia cysts, pollen, and ArP, yielding positive outcomes, particularly when fed on artemia cysts. Notably, artemia cysts resulted in a higher intrinsic rate of increase compared to natural prey, indicating their potential use as a substitute food for mass rearing *A.* eharai.

## Figures and Tables

**Figure 1 insects-14-00519-f001:**
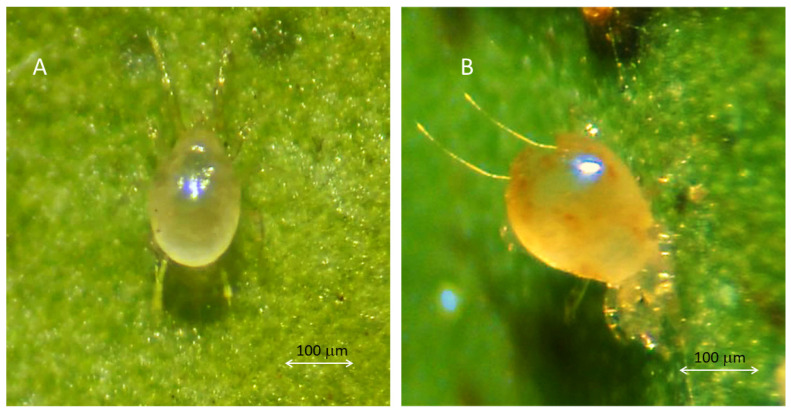
(**A**,**B**) Female of *Amblyseius eharai.*

**Figure 2 insects-14-00519-f002:**
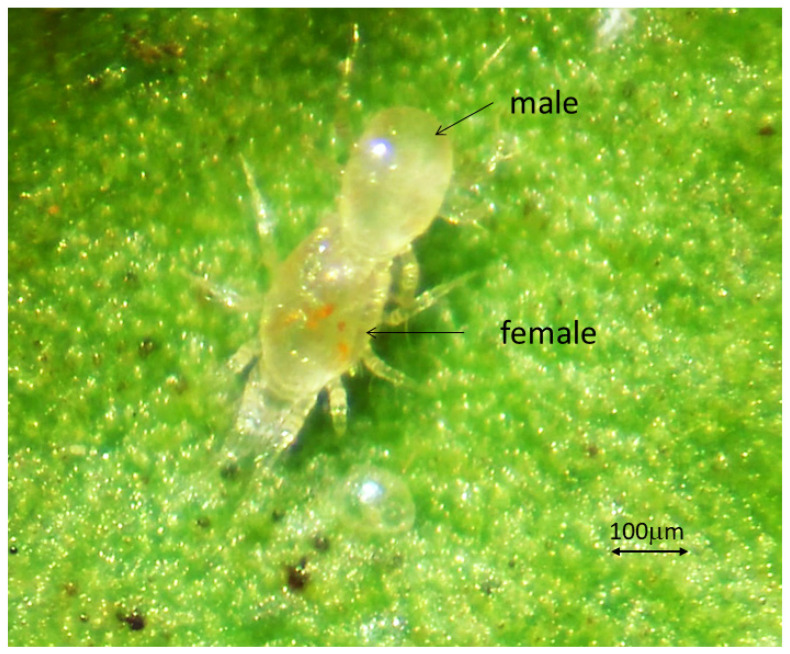
Two predatory mites (*Amblyseius eharai*) mating.

**Table 1 insects-14-00519-t001:** Experimental layout.

NT	R1	R2	R3
1	P.c	ArP	T.u
2	ArP	P.c	PH
3	A.f	T.u	ArP
4	T.u	PH	P.c
5	PH	A.f	A.f

NT: experiment; P.c: *Panonychus citri;* T.u: *Tetranychus urticae*; PH: pollen citrus; A.f: cysts of *A. Franciscana*; ArP: ArP artificial diet; R1, R2, R3: repetition 1, repetition 2, and repetition 3, respectively.

**Table 2 insects-14-00519-t002:** Development time (days) of *Amblyseius eharai* fed on different factitious and artificial foods.

Diet		Developmental Duration (Mean ± SE; Days)		
	*n*	Min	Max	Mean ± SE
*T. urticae*	30	7	9	8.20 ± 0.13 b
*P. citri*	30	5	7	6.90 ± 0.22 a
Artemia eggs	30	8	11	9.57 ± 0.19 c
Pollen	30	7	9	8.40 ± 0.25 b
Artificial diet	30	8	11	9.23 ± 0.21 c
F				25.29
df				4

Means within a column followed by the same letter are not significantly different (*p* > 0.05) according to a Tukey test; F, df, and *p* values refer to one-way ANOVAs.

**Table 3 insects-14-00519-t003:** Mean ± SE reproduction and longevity of *Amblyseius swirskii* on different factitious and artificial foods.

Diet	Preoviposition Period (Days) *	Oviposition Period (Days) *	Post-Oviposition Period (Days) *	Female Longevity (Days) *	Oviposition Rate (Eggs/Female/Day) *	Total Number of Eggs (Eggs/Female) *	Hatching Egg Proportion of the Progeny (%) *	Female Proportion of the Progeny (%) *
*T. urticae*	1.63± 0.11 a	21.27 ± 0.25 c	4.13 ± 0.29 a	35.23 ± 0.35 bc	1.89 ± 0.04 b	40.29 ± 0.83 bc	81.47 ± 0.71 a	63.77 ± 1.78 a
*P. citri*	2.60 ± 0.14 b	26.19 ± 0.46 d	6.30 ± 0.29 c	42.03 ± 0.43 c	1.74 ± 0.04 b	45.63 ± 0.94 c	78.38 ± 0.83 a	64.41 ± 1.29 a
Artemia cysts	2.67 ± 0.11 bc	18.03 ± 0.22 b	3.64 ± 0.19 a	33.93 ± 0.36 b	1.98 ± 0.04 c	35.87 ± 0.48 ab	80.72 ± 2.07 a	63.69 ± 1.21 a
Pollen	3.07 ± 0.15 cd	12.07 ± 0.35 a	5.24 ± 0.18 b	28.77 ± 0.24 a	1.71 ± 0.07 b	20.80 ± 1.14 a	79.95 ± 0.99 a	65.27 ± 1.75 a
Artificial diet	3.43 ± 0.10 d	13.13 ± 0.36 a	4.17± 0.19 a	30.97 ± 0.94 ab	1.48 ± 0.05 a	19.57 ± 0.77a	78.44 ± 1.29 a	60.75 ± 1.38 a
F	30.50	298.82	20.85	43.47	16.81	40.09	1.16	1.29
df	4	4	4	4	4	4	4	4
*p*	<0.001	<0.001	<0.001	<0.001	<0.001	<0.001	0.342	0.288

*: Means within a column followed by the same letter are not significantly different (*p* > 0.05). According to a Tukey test (preoviposition period, oviposition period, post-oviposition period, oviposition rate, hatching egg proportion of the progeny, female proportion of the progeny), according to a Rank test (female longevity, total number of eggs). F-, df-, and *p* values refer to one-way ANOVAs.

**Table 4 insects-14-00519-t004:** Life table parameters of *Amblyseius eharai* on different factitious and artificial food.

Diet	*n*	Net Reproductive Rate (R_0_. Females Per Female)	Generation Time (T. Days)	Intrinsic Rate of Increase (r_m_. Females/Female/Day)	Finite Rate of Population Increase (λ)
*T. urticae*	30	17.95	12.79	0.226	1.25
*P. citri*	30	23.06	14.21	0.221	1.24
Artemia cysts	30	19.13	12.18	0.242	1.27
Pollen	30	6.76	8.86	0.215	1.24
Artificial diet	30	7.54	10.28	0.196	1.22

## Data Availability

Data are available upon request from the corresponding author.

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
