# Peer review of "Effects of Different Diets on Biological Characteristics of Predatory Mite *Amblyseius Eharai* (Acari: Phytoseiidae)"

_insects, 2023, doi:10.3390/insects14060519_

Round 1

Reviewer 1 Report

Dear Authors,

The article is an interesting contribution about a phytoseiid considered a natural enemy of a reasonable group of prey. It is well written but needs to be rephrased, namely:

1. all species cited for the first time need author and family;

2. The introduction is complete and sufficient;

3. The methodology is well described, just not being clear about ANOVA and the comparison of means;

4. Results are highlighted correctly and sufficiently. However, the life table information is missing from the results.

5. The discussion is the disabled part. There should only be discussion and not data or results.

6. In my view, the article supports a conclusion. However, the conclusion presented is very simplistic. It is possible to better highlight the results obtained. In addition, it does not answer the research question.

Author Response

Dear reviewers,

We would like to express our sincere gratitude for your insightful feedback on our completed paper. We acknowledge and appreciate the suggestions provided by the reviewer, and we have made improvements to the paper accordingly. Here is a detailed account of the revisions we have made in response to the reviewer's comments:

We have added authors and family for the species initially cited in the paper.

We have included information about the lifespan in the Results section.

We have revised the Discussion and Conclusion sections.

We kindly request the reviewer to reevaluate our paper based on the aforementioned modifications.

Thank you for your time and consideration.

Best regards,

The author's team.

Reviewer 2 Report

The manuscript presents the Effects of five Diets on Biological Characteristics of Predatory Mite Amblyseius Eharai (Acari: Phytoseiidae).

The authors need to have a more careful reading and be clear in the materials and methods, objective in the results, and discussion.

Also, write a conclusion that reflects the results, looking at them in a holistic manner.

Several questions and proposals

Line

18

Write the diets

20-22

Change the sentence and include «the highest followed by A. Fanciscana cysts»

69

Environmental conditions are missing

88

Why do you need this operation? «Kill the embryos; then, the outer shells 89 were removed»

98

Effects of different diets on the development of Amblyseius eharai. The design is confused(a) Fifty separate rearing units were prepared, and an egg of A. eharai was placed in each 99 unit. 100 Each type of food was provided to ten rearing units. Is these 5 treatments and 10 replications? Why do you need: «The experiment was arranged in a completely randomized de-108 sign with five treatments and three replications»?

109

This is Table 1. Change the number of other tables

157

«A. eharai completed its life cycle in approximately 8.8 days, which is similar to the findings of this study» according to what temperature?

159

(Table 1) Either use two decimal places or one; standardize.

205

Discussion: The authors begin the discussion in the results by comparing with the work of other authors. I suggest "Results and Discussion" or to keep it as "Results" and include the discussion within the «Discussion» section.

The authors mention temperature conditions, but I couldn't find any text that presents the specific temperature at which the study was conducted.

281

Conclusion. This is not a conclusion and also is not exactly correct. In the objectives of the study, they only mention comparing five food substrates. However, looking at the table "Mean ± SE reproduction and longevity of Amblyseius swirskii on different factitious and artificial foods," the actual differences observed are: oviposition period (days), female longevity, and total number of eggs. The remaining differences are statistically significant but on a decimal scale. The best performance was observed with larval-stage T. urticae, followed by larval-stage P. citri, and then Artermia franciscana cysts. Please review the conclusion.

295

This table should stay before Conclusions

Author Response

Dear reviewers,

We would like to express our sincere appreciation for the profound feedback provided by the reviewer on our completed paper. We duly acknowledge and have made improvements to the paper based on the reviewer's suggestions. Here is a detailed explanation of the revisions we have made in response to the reviewer's comments:

We have corrected "the diets" in line 18.

We apologize for our oversight in not implementing this suggestion. We are still unclear about your opinion here, as the oviposition rate on the Artemia cysts diet was indeed the highest among all the diets. Could you kindly clarify your point further? We genuinely appreciate your input.

We used decapsulated Artemia cysts that had their hard outer shells removed and were not yet hatched. Therefore, killing the embryos was done to ensure that the Artemia cysts would not hatch during their usage in the study, even though this is a rare occurrence. Additionally, based on our observations, decapsulated Artemia cysts that underwent embryo termination preserved better in the refrigerator compared to those that were not subjected to embryo termination.

The experimental design in our study involved using 50 rearing units (50 individual ) for each repetition, this mean 10 individual per diet tested in each repetition. This means that after three repetitions, the total number of experimental specimens for each diet was 30 individuals. This was done to ensure an adequate sample size in the experiment. Since each rearing unit only contained one individual, we did not consider one rearing unit as one repetition.

We have rearranged the table numbering as per your suggestion.

We have added the temperature information. The conditions in our laboratory were as follows: at 25 ± 1°C and 75 ± 5% relative humidity with a 16L:8D light-dark cycle (line 79).

We have used two decimal places for all values in Table 1.

The Discussion and Conclusion sections have been improved.

We kindly request the reviewer to reevaluate our paper in light of these revisions.

Thank you for your consideration and valuable feedback.

Best regards,

The author's team.

Round 2

Reviewer 1 Report

The Article is ready for publication.

Author Response

Dear reviewers,

We would like to express our sincere gratitude for accepting our revised paper. Your acceptance means a great deal to us, and we truly appreciate it.

Once again, thank you very much, and we wish you good health.

Best regards,

The author's team.

Reviewer 2 Report

Ok for me although not all questions have been answered
Minor revisions

Put Table 4 before the discussion.

Supplementary file: please use two decimal places in every number.

Author Response

Dear reviewers,

First and foremost, we would like to sincerely apologize for the oversight on our part during the first round of revisions, which resulted in missing some of your suggestions. We deeply regret that this oversight led to an imperfect experience with our initial revised submission.

In this second round of revisions, we have addressed these shortcomings by:

Moving Table 4 ahead of the Discussion section.

Adding two decimal places for the Supplementary file.

We truly appreciate your thorough and meticulous feedback, which has contributed to the improvement of our paper.

Thank you.

Best regards,

The author's team.